# Simplicity is Key: An Unsupervised Pretraining Approach for Sparse Radio Channels

## Abstract

We introduce **Spa**rse pretrained **R**adio **Tran**sformer (SpaRTran), an unsupervised representation learning approach based on the concept of compressed sensing for wireless channels. SpaRTran learns embeddings that focus on the physical properties of radio propagation to allow for an efficient fine-tuning on radio-based downstream tasks. SpaRTran uses a sparse gated autoencoder that induces a simplicity bias in the learned representations, resembling the sparse nature of radio propagation. For signal reconstruction, it learns a dictionary that holds atomic features, which increases flexibility across signal waveforms and spatio-temporal signal patterns. Compared to the state of the art, SpaRTran cuts positioning error by up to 28 % and increases top-1 codebook selection accuracy for beamforming by 26%pts. By pretraining models solely on individual channel measurements, it is system-agnostic and more versatile, allowing fine-tuning for diverse radio tasks and substantially reducing labeling costs.

## 1 Introduction

Wireless communication networks continuously capture channel state information (CSI), i.e. estimates of the channels between transmitters and receivers. Physically, the CSI describes how the wireless medium, i.e. the physical and geometrical properties of the environment, transforms (scales, delays, phase-rotates and Doppler-shifts) the transmitted electromagnetic waveforms before they reach the receivers. In the past, this information was used to equalize the channel influence minimizing inter symbol interference and signal distortion. Today, the channel is reinterpreted as a source of spatio-temporal information rather than an impairment leading to new spatially-aware applications such as beamforming (steering signal energy into a specific direction to improve reception in a targeted area) or wireless positioning. This development is driven by the rise of deep learning algorithms capable of extracting complex pattern from the channel (Zhang et al., 2019). However, acquiring labels to train neural networks in an purely supervised manner is labor-intensive and the dynamic nature of environments requires a constant retraining of the networks to maintain high performance Stahlke et al. (2022).

Unsupervised learning has shown significant improvements in domains such as natural language processing (Devlin et al., 2019; Radford et al., 2018) and computer vision (Grill et al., 2020; Caron et al., 2021; He et al., 2020; Chen et al., 2020), often requiring fewer labeled samples for fine-tuning. Hence, the paradigm has been applied to train foundation models for wireless channels that can be efficiently finetuned on various downstream tasks, achieving state-of-the-art accuracy with significantly less labeled data (Salihu et al., 2024; Ott et al., 2024; Alikhani et al., 2024; Pan et al., 2025). However, existing approaches still face two key challenges: First, to maximize versatility, general-purpose wireless foundation models should be pretrained on single-channel measurements that generalize across system setups and downstream tasks, rather than on full CSI that typically collects channel measurements from every available antenna, making it specific to a particular system configuration. Second, core assumptions of prominent self-supervised learning (SSL) methods often misalign with CSI. For instance, in vision, separating class representations is sensible, but CSI measurements vary smoothly across space and thus exhibit subtler relationships (Studer et al., 2018).

We propose Sparse pretrained Radio Transformer (SpaRTran), which introduces inductive biases by embedding physical knowledge into both the model architecture and the training process, thereby

improving training efficiency and the quality of the learned representations. Rather than adapting existing SSL methods, we propose a novel, purely unsupervised pretraining approach specifically tailored for CSI. Our physics-informed method draws inspiration from the concept of sparse autoencoder (SAE) (Lee et al., 2007) and compressed sensing (CS) (Donoho & Huo, 2001; Candes et al., 2006). The central premise of CS is that sufficiently sparse representations reduce ambiguity; in contrast, non-sparse representations typically contain numerous insignificant components, complicating both analysis and signal recovery (Donoho & Huo, 2001). Our contributions are three-fold: (1) We propose a novel sparsity based pretraining framework that maps the input signals into high-dimensional sparse vectors and then reconstructs the original signal, conforming to the signal properties caused by the physical radio signal transmissions. (2) We use a learned dictionary to obtain sparse signal representations while maintaining flexibility in the employed signal waveforms. (3) We build on a gated SAE (Rajamanoharan et al., 2024) using a transformer (TF) neural network as backbone to learn sparse representations while preserving reconstruction fidelity. We extend it by a phase generator network that integrates complex phase information into the sparse signal coefficients, thereby enabling the representation of signal phases through complex-valued components while maintaining sparsity via the gating mechanism.

## 2 RELATED WORK

SpaRTran is an unsupervised representation learning method for pretraining on wireless signals that integrates techniques from compressed sensing and dictionary learning.

**Unsupervised pretraining for wireless channels.** In recent years, supervised deep learning revolutionized tasks such as wireless positioning (Salihu et al., 2022; Liu et al., 2022; Zhang et al., 2023) and beam-management (Ma et al., 2023). Today, unsupervised learning and SSL attracts high attention in this context. The aim is to leverage cost effective unlabeled channel measurements to pre-train a task agnostic basis model also known as foundation model. Existing works transfer the pretraining objectives that have been established in domains such as computer vision or natural language processing to the wireless domain. Here, contrastive (Salihu et al., 2024) and predictive Alikhani et al. (2024); Catak et al. (2025); Ott et al. (2024); Yang et al. (2025) methods have been studied as well as their combination (Pan et al., 2025; Guler et al., 2025). However, although most of these methods include a preprocessing stage that accounts for the specific properties of channel measurements, their pre-training objectives (such as recovering masked inputs or pulling similar samples closer together in embedding space) are developed for substantially different data domains. This raises the question of whether an objective that takes the unique physical properties of wireless signals into account can improve performance over existing methods.

**Compressed sensing** represents signals by a high-dimensional sparse vector in an overcomplete basis, assuming they arise from few latent factors. Common applications of CS in wireless systems include wireless source separation (Donoho, 2006; Candes et al., 2006), direction-of-arrival (DOA) estimation (Yang et al., 2018), and channel estimation (Berger et al., 2010). Basis pursuit denoising (Chen et al., 2001; Tibshirani, 1996) uses convex relaxation to turn nonconvex sparse recovery into a convex problem, while greedy methods like orthogonal matching pursuit (OMP) (Tropp & Gilbert, 2007) iteratively select active atoms; sparse Bayesian learning enforces sparsity via probabilistic priors (Malioutov et al., 2005; Stoica et al., 2011). Recent deep-learning-based CS methods focus on the compression of the signals reducing complexity and improving reconstruction fidelity compared to conventional approaches (Machidon & Pejović, 2023). Another direction of research are SAEs (Cunningham et al., 2023; Bricken et al., 2023). SAEs enforce sparsity via regularization in high-dimensional latent spaces, yielding more interpretable features than bottleneck autoencoders. Inspired by gated linear units (Dauphin et al., 2017), Rajamanoharan et al. (2024) address low reconstruction accuracy resulting from biases introduced by the sparsity constraint by decoupling the selection of active components from the estimation of sparse coefficients. To the best of our knowledge, SpaRTran is the first to apply CS to the design of unsupervised pretraining.

**Dictionary learning** algorithms identify atomic features that sparsely represent underlying data, i.e., the dictionary is learned empirically from the signals themselves. This enables generalization across signal types and often leads to increased sparsity (Elad, 2010). A prominent example is the K-SVD algorithm (Aharon et al., 2006), that iteratively updates the dictionary atoms. Rather than

using a fixed theoretical dictionary, SparTRan can jointly learn the dictionary, increasing flexibility across waveforms and spatio-temporal patterns.

# 3 PROBLEM DESCRIPTION

During a radio signal transmission, the electromagnetic wave interacts with the environment, i.e., the channel, which affects the signal, resulting in multiple propagation paths arriving at the receiver. The received signal $y(t)$ can be defined as $y(t) = h(t) * s(t) + w(t)$, where $s(t)$ is the transmitted signal, $h(t)$ the channel, $w(t)$ additive white Gaussian noise and $*$ the convolution operator. The channel impulse response (CIR) $h(t)$ characterizes the radio transmission channel and can be modeled as

$$h(t) = \sum_{k=0}^{K-1} \alpha_k e^{-i\varphi_k} \delta(t - \tau_k), \tag{1}$$

where $\tau_k$ is the signal transmission delay, $\alpha_k$ the magnitude and $\varphi_k$ the phase of the $k$-th propagation path of the transmitted signal. $\delta$ denotes the Dirac delta function and $i$ the imaginary unit. Eq. 1 is the superposition of several signals, originating from $K$ far field sources. In practice, $K$ is assumed to be unknown. The bandwidth-limited discrete channel measurement is modeled as

$$h[m] = \sum_{k=0}^{K-1} a_k \text{sinc}[m - \tau_k W] + w_m, \tag{2}$$

where $W$ is the bandwidth of the system, $a_k$ is the complex valued path coefficients, and $m \in \{1, \cdots, M\}$. From this, we derive the sparse channel representation. Assuming a set of $L$ potential signals $\psi_l \in \mathbb{R}^M$ that form a basis, of which only $K \ll L$ effectively contribute to the received signal, we can rewrite Eq. 2 as

$$\boldsymbol{h} = \sum_{l=0}^{L-1} a_l \psi_l + \boldsymbol{w}, \tag{3}$$

where $|\alpha_l| > 0$ if the $l$-th signal is an active signal component, and $|\alpha_l| = 0$ otherwise. Note that we have replaced the sinc-function with a more generic notation $\psi_l$. Defining the dictionary $\boldsymbol{\Psi} := [\psi_0, \cdots, \psi_{L-1}]$ allows (3) to be expressed more concise in matrix notation as

$$\boldsymbol{h} = \boldsymbol{\Psi}\boldsymbol{a} + \boldsymbol{w}, \tag{4}$$

where $\boldsymbol{a} = [\alpha_0, \cdots, \alpha_{L-1}]^T$ is the sparse coefficient vector, and $\boldsymbol{\Psi}$ is a $M \times L$ dictionary matrix. Eq. 4 describes an underdetermined system of equations. As there is no unique solution, recovering the sparse channel requires solving the following optimization problem:

$$\min \|\boldsymbol{a}\|_0, \quad \text{s.t. } \|\boldsymbol{\Psi}\boldsymbol{a} - \boldsymbol{h}\|_2 \leq \epsilon, \tag{5}$$

where $\epsilon$ denotes the allowed reconstruction error due to noise. Eqs. (4) and (5) together describe the radio channel within the framework of compressed sensing (Donoho, 2006; Candes et al., 2006).

## 3.1 THEORETICAL ANALYSIS

We treat the incoming signal as a time-dependent function $f(t)$ to be expressed in a reproducing kernel Hilbert space (RKHS) $\mathcal{H}$ with bases $\{\varphi_i\}_{i=0}^N$. For the sake of theoretical analysis, we analyze an invertible operator $O : \mathcal{H} \rightarrow \mathcal{H}$ mapping from measured functions $f \in \mathcal{H}$ to latent representations $\tilde{f} = O[f]$. Our idea is to model the $\tilde{f}$ rather than $f$ itself to reach a more compact representation. We will first characterize the error of a signal transformed by an invertible operator $O$. Note that all proofs for the theorems can be found in the appendix.

**Theorem 1.** *Let $\mathcal{H}$ be a reproducing kernel Hilbert space, equipped with a basis $\{\varphi_i\}_{i=0}^N$. For any $f \in \mathcal{H}$ let the best n-term approximator be*

$$\sigma_n(f) = \min_{|I| \leq n} \|f - \sum_{i \in I} a_i \varphi_i\|_{\mathcal{H}}.$$

*Also define the 1-atomic norm as*

$$\|f\|_{A_1(H)} = \inf\{\sum_i \|a_i\| : f = \sum_i a_i \varphi_i\}$$

*Assume there exists an exact recovery condition (ERC (Tropp, 2004), the $O(n^{1/2})$ rate is optimal (Klusowski & Siegel, 2025)[1]) a constant $C > 0$ such that for every $f \in \mathcal{H}$ we have*

$$\sigma_n(f) \leq C \frac{\|f\|_{A_1(\mathcal{H})}}{\sqrt{n}}$$

*Let $O : \mathcal{H} \to \mathcal{H}$ be any invertible bounded linear operator with the standard operator norm $\|O\| = \sup_{\|f\|_{\mathcal{H}}=1} \|O[f]\|_{\mathcal{H}}$. Define the O-atomic norm of $f \in \mathcal{H}$ as*

$$\|f\|_{A_1^O} = \|O[f]\|_{A_1(\mathcal{H})}$$

*Then there exists an n-term representation $g_n$ in $\mathcal{H}$ such that*

$$\|O[f] - g_n\|_{\mathcal{H}} \leq C \frac{\|f\|_{A_1^O}}{\sqrt{n}}$$

*or, in the original space: $g_n = O[\tilde{f}_n]$*

$$\|f - \tilde{f}_n\|_{\mathcal{H}} \leq \|O^{-1}\| C \frac{\|f\|_{A_1^O}}{\sqrt{n}}$$

The result in Theorem 1 implies that if one tries to bound $K$ functions simultaneously,

$$\max_{0 \leq i \leq K} \|f_i - f_{i,n}\| \leq \|O^{-1}\| C \frac{\max_{0 \leq i \leq K} \|f_i\|_{A_1^O}}{\sqrt{n}}$$

preconditioning with $O$ improves upon the direct estimation if there exists a common substructure that allows us to reduce the coefficients in $\|f_i\|_{A_1^O}$ without overly affecting $\|O^{-1}\|$.

This is not unexpected: This is exactly what happens in signal smoothing where $O$ flattens singularities, or in differential preconditioning for PDEs. We provide an example of an analytically derived operator in Appendix B. Instead of using prior knowledge, we decide to learn $O$ by fitting it jointly with its dictionary $\Psi$ on a dataset of $\{f_0, \ldots, f_K\}$ such that the operator preconditions the entire set.

Specifically, we can construct such an operator the following way

**Theorem 2.** *Let $\{f_i\}_{i=0}^K$ be a datset of signals, and $\mathcal{H}$ be a hilbert space with a (possibly infinite) basis $\{\varphi\}_{j=0}^N$. Let $R = \max_{0 \leq i \leq K} \sum_{j=1}^N |a_{i,j}|$ and fix a nonempty index set $S \subset \{1, \ldots, N\}$. Define*

$$R_S := \max_{0 \leq i \leq K} \sum_{j \in S} |a_{i,j}| \qquad B := \max_{0 \leq i \leq K} \sum_{j \notin S} |a_{i,j}|$$

*If $B < R_S$ then there exists an operator $O$ such that*

$$\max_{0 \leq i \leq K} \|f_i - f_{i,n}\| \leq C \frac{\max_{0 \leq i \leq K} \|f_i\|_{A_1^O}}{\sqrt{n}}$$

*which is strictly better than the original rate $C \frac{R}{\sqrt{n}}$*

**Remark 1.** *The assumptions in Theorem 2 are quite technical, but can be boiled down to "One can sacrifice a more irrelevant subset $\bar{S}$ in favor of modeling a relevant subset $S$ with higher weight." In practice this is a small assumption for an uninformed selection of the basis.*

Of course the "small index set" assumption for Theorem 2 is generally restrictive, but we can easily generalize the proof towards a low-rank assumption:

**Corollary 1.** *Assume there exists a rank $s$ subspace $S$, then the same bounds from theorem 2 hold.*

One important consequence of Corollary 1 is that we can find better sparse representations by finding a dense rotation and diagonal scaling. While simple finite operations at first appear to be sufficient for this, one has to note that the basis set we work with is generally infinite, meaning an orthogonal projection is not naively parameterizable. We decide to approximate $O$ with a TF network by jointly minimizing the reconstruction error after applying $O$ together with the codebook $\{\varphi\}$.

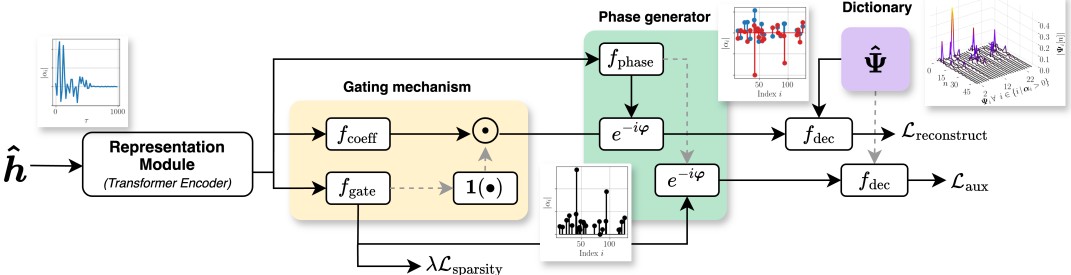

Figure 1: Overview of our unsupervised pretraining method - SparTRan.

## 4 SPARTRAN TRAINING PIPELINE

In general, we consider a set of unlabeled channel measurements $\mathcal{M}$. Our objective is to learn channel representations that encode the environmental characteristics of the radio signal. To this end, we introduce a strong sparsity bias into the training process through both model architecture and loss function design. Our approach employs an encoder that generates a latent representation $z$ and a decoder that reconstructs the input signal $\hat{h} \sim \mathcal{M}$ based on $z$.

A TF architecture (Vaswani et al., 2017) forms the backbone of the encoder. We employ a lightweight encoder-only-TF with a depth of one and internal latent dimension of $N_{latent} = 512$ using 8 attention heads for the multihead attention mechanism. We use a three-dimensional vector $\tilde{h}_m$, consisting of the real, imaginary and absolute parts of the complex number, to represent the complex values at the $m$-th timestep $\hat{h}_m$. To construct the input embedding $e$ we combine windows of three time steps of the CIR to an input token $e_m = [\tilde{h}_{3m-2}, \tilde{h}_{3m-1}, \tilde{h}_{3m}]^T$. We project each input token into the latent space of dimension $N_{latent}$ via a learned linear transformation to match the internal dimensionality of the TF-encoder.

### 4.1 SPARSE RECONSTRUCTION HEAD

The sparse reconstruction head uses a gating mechanism (inspired by Rajamanoharan et al. (2024)) and a phase generator. The former promotes the reconstruction to be sparse while the latter converts the real numbered output of the neural network to the complex valued coefficients $\hat{a}$. $\hat{a}$ represents the reconstructed signal in terms of a overdetermined dictionary $\Psi$, see Eq. 4. Fig. 1 shows the gating mechanism (yellow), the phase generator (green), and the dictionary (purple).

We now discuss the gating mechanism in more detail. Approximating the $l_0$-norm with the $l_1$-norm tends to lead to a non-optimal reconstruction. This is due to the fact that the sparsity penalty, i.e., the $l_1$-norm, can be reduced at the cost of reconstruction performance (Wright & Sharkey, 2024). Hence, our strategy for the estimation of $\hat{x}$ follows the work of Rajamanoharan et al. (2024). The idea is to separately handle the selection of active atoms from the dictionary ($f_{\text{gate}}$) and the estimation of the coefficients magnitude ($f_{\text{coeff}}$). The encoder output is defined by

$$\hat{x} = f_{\text{coeff}}(z) \odot \mathbf{1}(\underbrace{f_{\text{gate}}(z)}_{\rho_{\text{gate}}}), \tag{6}$$

where $\mathbf{1}$ denotes the Heaviside step function, $\odot$ the Hadamard product and $\rho_{\text{gate}}$ is the output of the gating stage before the binarization step. Fig. 1 shows the gating mechanism (yellow). Due to the binarization of the gating values, no gradient flows through this path of the network, see grey arrows in the yellow box of Fig. 1. Thus, an auxiliary loss promotes the detection of active atoms in $f_{\text{gate}}$. The auxiliary loss measures reconstruction fidelity, but instead of $\hat{x}$, it uses $\rho_{\text{gate}}$ to reconstruct the signal. The dictionary should not be updated by the auxiliary reconstruction task. Hence, we prohibit the flow of the gradient accordingly (see grey dashed line in Fig. 1).

We now outline our extensions to the original method. Rajamanoharan et al. (2024) restrict the encoders output $\hat{x}$ to real positive numbers. However, this assumption does not hold in our case, as our

---

[1]This is equivalent to asserting a generalised jackson-type inequality over the codebook established in (Temlyakov, 2011)

goal is to estimate complex-valued path coefficients $\hat{a}$. To address this, we interpret the outputs of $f_{\text{coeff}}$ and $f_{\text{gate}}$ as the magnitudes of the complex coefficients. This formulation allows us to suppress negative values via the gating mechanism without violating the underlying physical channel model. In addition, we introduce a third function $f_{\text{phase}}$, that generates the phases of the path coefficients. The final coefficients are then constructed as: $\hat{a} = \hat{x}e^{-if_{\text{phase}}(z)}$ and $\rho'_{\text{gate}} = \rho_{\text{gate}}e^{-if_{\text{phase}}(z)}$, where $i$ denotes the imaginary unit. The output of $f_{\text{phase}}$ is constrained to the interval $\pm\pi$ using a scaled tanh activation function. This leads to the following loss function:

$$\mathcal{L} := \underbrace{\|\tilde{h} - f_{\text{dec}}(\hat{a}, \hat{\Psi})\|_2^2}_{\text{reconstruction loss}} + \underbrace{\lambda\|\mathbf{1}(\rho_{\text{gate}})\|_1}_{\text{sparsity penalty}} + \underbrace{\|\tilde{h} - f_{\text{dec}}(\rho'_{\text{gate}}, \hat{\Psi}_{\text{frozen}})\|_2^2}_{\text{auxiliary loss}}, \qquad (7)$$

with $f_{dec}(\hat{a}, \hat{\Psi}) = \hat{\Psi}\hat{a}$. To enforce non-negativity, Rajamanoharan et al. (2024) employ ReLU activations for $f_{\text{gate}}$ and $f_{\text{coeff}}$. We observed that this can lead to a situation where certain dictionary atoms are never activated, i.e., their associated coefficients remain zero, resulting in no gradient updates, a phenomenon akin to the dying ReLU problem. To mitigate this, we use leaky ReLU activations (slope 0.01), ensuring that gradients can still propagate even for inactive units.

## 4.2 DICTIONARY LEARNING

If crucial system parameter, such as signal bandwidth or the receivers sampling frequency, are unknown, the construction of a dictionary that conforms to the theory is unfeasible. This may occur when using diverse crowd-sourced signals to train a large foundation model. Hence, instead of using a fixed dictionary that conforms to the theoretical channel model (Sec. 3), SpaRTran can treat the dictionary as a learnable parameter $\hat{\Psi}$, see Fig. 1, purple box. By, normalizing the atomic entries to unit norm, they only determine the direction of the contribution, while $\hat{a}$ provides the amplitude and phase of the complex-valued signal component. This approach enables the model to learn more expressive atoms that capture complex interactions, adapt to diverse wireless pulse shapes, and increases dictionary incoherence—improving the ability to identify contributing atoms Donoho & Huo (2001). However, jointly optimizing an unrestricted dictionary renders problem (5) NP-hard Tropp (2004).

## 4.3 FINETUNING

While SpaRTran learns representations for individual transmitter–receiver links, most tasks exploit correlations across multiple channels. We consider $N_r$ links recorded by an agent traversing the environment. We aggregate these into the channel state $H = [\hat{h}_1 \cdots \hat{h}_{N_r}]$, denoted the CSI. Hence, for finetuning, we compute a representation for each available link and concatenate them to form a complete representation of the CSI. We use small ResNet as finetuning head. It employs three basic blocks with 12, 32, and 64 channels consisting of two sequential 1D convolutional layers and a residual connection. The weight sharing of the convolutional layers circumvents an unreasonable increase in parameters when $N_r$ is large.

## 5 EVALUATION

We evaluate localization via CSI fingerprinting and codebook selection for beamforming — both leveraging full CSI complexity. For our experiments we compare three SSL baselines (**SWiT** (Salihu

Table 1: Fingerprinting (FP) performance for SparTRan and the baselines models for different amounts of labeled training data evaluated on the FH-IIS dataset (MAE / CE90 in meter).

| Method | 1% | 2% | 5% | 10% | 25% | 50% | 100% |
|---|---|---|---|---|---|---|---|
| **Masking** | **0.73 / 1.27** | 0.61 / 1.08 | **0.50 / 0.89** | 0.48 / 0.84 | 0.42 / 0.75 | 0.44 / 0.76 | 0.40 / 0.70 |
| **LWM** | 1.17 / 2.10 | 0.91 / 1.63 | 0.72 / 1.27 | 0.63 / 1.11 | 0.56 / 0.99 | 0.55 / 0.97 | 0.58 / 0.99 |
| **SWiT** | 2.33 / 4.24 | 2.09 / 3.81 | 1.90 / 3.46 | 1.79 / 3.29 | 1.76 / 3.22 | 1.65 / 3.01 | 1.51 / 2.75 |
| **SpaRTran** | 0.77 / 1.40 | **0.58 / 1.04** | **0.50 / 0.91** | 0.41 / **0.74** | **0.34 / 0.62** | **0.30 / 0.56** | **0.30 / 0.55** |
| **WiT (Sup.)** | 1.27 / 2.32 | 1.05 / 1.88 | 0.88 / 1.59 | 0.89 / 1.61 | 0.66 / 1.21 | 0.56 / 1.03 | 0.49 / 0.90 |

Table 2: FP performance across different system setups finetuned on $5\,000$ samples of the KUL dataset (MAE / CE90 in meter).

| Method | Pretrain-Set | DIS-LoS | ULA-LoS | URA-LoS | URA-nLoS |
|---|---|---|---|---|---|
| **Masking:** | DIS-LoS | 0.093 / 0.158 | 0.065 / 0.118 | 0.071 / 0.129 | 1.176 / 1.626 |
| | ULA-LoS | 0.081 / 0.139 | 0.067 / 0.116 | 0.071 / 0.127 | 1.094 / 1.615 |
| | URA-LoS | 0.087 / 0.156 | 0.068 / 0.118 | 0.073 / 0.131 | 1.176 / 1.671 |
| | URA-nLoS | 0.072 / 0.128 | 0.067 / 0.120 | 0.073 / 0.139 | 1.153 / 1.627 |
| **SWiT:** | DIS-LoS | **0.071** / 0.139 | 0.069 / 0.138 | 0.057 / 0.119 | 0.154 / 0.324 |
| | ULA-LoS | 0.076 / 0.146 | 0.068 / 0.136 | 0.058 / 0.119 | 0.140 / 0.303 |
| | URA-LoS | **0.070** / 0.138 | 0.068 / 0.136 | 0.057 / 0.119 | 0.156 / 0.327 |
| | URA-nLoS | 0.077 / 0.148 | 0.068 / 0.137 | 0.059 / 0.119 | 0.152 / 0.326 |
| **LWM:** | DIS-LoS | 0.083 / 0.144 | 0.084 / 0.151 | 0.075 / 0.137 | 0.244 / 0.454 |
| | ULA-LoS | 0.085 / 0.151 | 0.082 / 0.146 | 0.079 / 0.147 | 0.256 / 0.500 |
| | URA-LoS | 0.075 / 0.132 | 0.082 / 0.147 | 0.065 / 0.122 | 0.204 / 0.372 |
| | URA-nLoS | 0.094 / 0.164 | 0.092 / 0.163 | 0.083 / 0.151 | 0.224 / 0.418 |
| **SpaRTran:** | DIS-LoS | 0.127 / 0.186 | **0.055** / **0.098** | **0.043** / **0.087** | **0.071** / **0.131** |
| | ULA-LoS | **0.065** / **0.114** | **0.058** / **0.100** | **0.054** / **0.110** | **0.075** / **0.139** |
| | URA-LoS | 0.072 / **0.122** | **0.048** / **0.085** | **0.051** / **0.100** | **0.080** / **0.145** |
| | URA-nLoS | **0.064** / **0.109** | **0.048** / **0.082** | **0.038** / **0.073** | **0.077** / **0.142** |
| **WiT (Sup.):** | | 0.074 / **0.135** | 0.071 / 0.135 | 0.059 / 0.108 | 0.196 / 0.387 |

Table 3: Top-1 accuracy (%) for codebook selection in the beamforming task. Fine-tuning was performed across task complexities defined by codebook sizes (16, 32, 64, 128) and varying proportions of labeled training data (1%, 2%, 5%, 10%, 25%, 50%, 100%).

| Method | Codebook Size | 1% | 2% | 5% | 10% | 25% | 50% | 100% |
|---|---|---|---|---|---|---|---|---|
| **Masking** | 16 | 50.7 | 59.6 | 68.5 | 72.4 | 76.8 | 78.4 | 80.6 |
| | 32 | 31.6 | 40.6 | 53.5 | 67.6 | 74.9 | 78.1 | 81.2 |
| | 64 | 15.9 | 23.4 | 33.2 | 43.6 | 59.1 | 64.8 | 65.0 |
| | 128 | 9.4 | 10.1 | 13.0 | 20.8 | 33.0 | 40.4 | 42.1 |
| **SWiT** | 16 | **56.7** | 61.3 | 73.5 | 78.6 | 81.3 | 83.7 | 84.3 |
| | 32 | 40.0 | 50.6 | 71.1 | 76.4 | 77.2 | 80.0 | 81.5 |
| | 64 | 30.1 | 36.7 | 51.6 | 60.1 | 69.5 | 70.2 | 68.0 |
| | 128 | 18.6 | 22.6 | 25.3 | 36.5 | 44.6 | 47.6 | 41.2 |
| **LWM** | 16 | 40.3 | 51.0 | 68.7 | 72.5 | 77.4 | 79.1 | 81.2 |
| | 32 | 21.0 | 28.4 | 51.9 | 67.3 | 76.3 | 82.0 | 82.4 |
| | 64 | 6.8 | 12.5 | 38.7 | 61.3 | 72.2 | 67.8 | 69.4 |
| | 128 | 4.0 | 4.3 | 10.8 | 17.5 | 51.8 | 44.0 | 57.7 |
| **SpaRTran** | 16 | 56.0 | **71.6** | **82.8** | **84.6** | **89.4** | **90.0** | **92.8** |
| | 32 | 38.7 | **73.6** | **87.6** | **88.9** | **90.4** | **92.9** | **93.7** |
| | 64 | **32.3** | **54.2** | **74.4** | **84.2** | **86.6** | **89.5** | **89.0** |
| | 128 | **23.8** | **25.7** | **39.9** | **62.9** | **70.5** | **74.5** | **71.8** |
| **WiT (Sup.)** | 16 | 56.4 | 67.3 | 74.3 | 80.1 | 82.3 | 84.3 | 85.8 |
| | 32 | **44.3** | 69.1 | 70.2 | 79.4 | 82.1 | 82.9 | 84.4 |
| | 64 | 31.3 | 49.0 | 52.3 | 60.4 | 73.3 | 75.5 | 77.6 |
| | 128 | 15.9 | 24.2 | 21.7 | 34.4 | 48.4 | 48.8 | 49.8 |

et al., 2024), **Masking** (Ott et al., 2024), and **LWM** (Alikhani et al., 2024)) and a supervised method (**WiT** (Salihu et al., 2022)), each using a TF backbone. We evaluate all methods on three publicly available datasets: (i) **KUL**, a small controlled environment (Bast et al., 2020), (ii) **FH-IIS**, a larger and more complex environment (Stahlke et al., 2024), and (iii) **DeepMIMO** (Alkhateeb, 2019) for large-scale, diverse urban scenarios used in the codebook selection task. More detailed information on the baselines and the datasets and how we parameterized them can be found in Appendix C.

**Downstream Tasks.** We use two downstream tasks to evaluate the performance of SpaRTran:
(1) *Radio fingerprinting* estimates a position in a known environment by exploiting the high spatial correlation of channel measurements. Due to multipath the CSI is typically highly characteristic per position. Assuming a wide-sense static environment (i.e., negligible changes in the radio environ-

ment between training and inference) a neural network can be trained to map these fingerprints to positions for localization (Niitsoo et al., 2019; Stahlke et al., 2022), see Secs. 5.1 and 5.2.

(2) *Beamforming* adapts phases/amplitudes across large phased array antennas to perform directed signal transmissions, mitigating high path losses and interference. Codebook selection for beamforming aims to select the optimal beam from a predefined codebook directly from channel measurements. This reduces the channel estimation overhead (Giordani et al., 2019), see Sec. 5.3.

In Sec. 5.4 we evaluate the approach under several ablations and study the effect of varying values for sparsity penalty $\lambda$ and dictionary size $L$,

## 5.1 LOCALIZATION IN SMALL DATA REGIME

Table 1 shows the mean absolute error (MAE) and 90th percentile of the cumulative error (CE90) of SparTRan and the baseline methods on the FH-IIS dataset. SpaRTran offers in most cases the highest accuracy, demonstrating the effectiveness of the approach. With $\geq 25\%$ of the training data available, SpaRTran reduces the average MAE and CE90 relative to the best-performing baseline ("Masking") by 23% and 19% respectively. SpaRTran significantly outperforms the purely supervised approach WiT — an average improvement of $0.67\,\mathrm{m}$ CE90 and $0.37\,\mathrm{m}$ MAE — demonstrating the advantage of unsupervised pretraining. Given very few labeled datapoints ($\leq 5\%$ of dataset) SpaRTran performs competitive to Masking but does not reliably outperform it. We attribute this to the fact that SpaRTran is trained on single channels rather than the full CSI requiring it to learn inter channel correlations solely during finetuning. SWiT's much lower accuracy compared with the supervised case suggests its rigid pretraining augmentations mismatch the target data. This underscores the need for pretraining methods tailored to wireless-signal properties that remain flexible for different system configurations.

## 5.2 LOCALIZATION UNDER DOMAIN SHIFT

Table 2 presents the results of wireless localization trained on pairs of scenarios one used for pretraining the other for finetuning. SpaRTran achieves the best accuracy in most cases. Across all training pairs, SpaRTran achieves an average improvement of at least $19\,\%$ in MAE and $28\,\%$ in CE90 and reaching MAE $\leq 0.080\,\mathrm{m}$ and CE90 $\leq 0.145\,\mathrm{m}$ even in the challenging URA-nLoS case. While all baseline methods exhibit a marked decline in performance under challenging non-line-of-sight (nLoS) conditions compared to line-of-sight (LoS) scenarios, SpaRTran maintains a high accuracy. This highlights SpaRTran's superior ability to extract meaningful signal features beyond the dominant LoS path. It should be noted that in the simple LoS scenarios the purely spervised WiT (Sup.) shows competitive performance to the unsupervised approaches, reaching MAE $\leq 0.074\,\mathrm{m}$ and CE90 $\leq 0.135\,\mathrm{m}$. This shows that pretrained methods offer the most benefit in particular complex situations, whereas supervised approaches may perform adequately in simpler settings even with limited training data.

## 5.3 BEAMFORMING

Table 3 shows the results for the beamforming downstream task, i.e., selecting the best beam id in a predefined codebook to steer antenna gain in a specific direction to serve a mobile device. Again, SpaRTran outperforms the compared methods consistently. Notably, in one of the most challenging settings — a codebook size of 128 with fine-tuning on only 10 % of the labeled training data — SpaRTran increases top-1 accuracy by 26 percentage points up to 62.7 % relative to SWiT. It is noteworthy that in the general picture none of the self-supervised baselines significantly outperforms the purely supervised method WiT, supporting our view that simply adapting existing pretraining methods to the wireless domain is insufficient. With only 1% labeled data, SpaRTran's advantage diminishes. This likely reflects that it was pretrained on single-channel measurements rather than full CSI matrices, so fine-tuning must learn inter-channel correlations from very few labeled examples and thus needs more labeled data to show its full benefit.

## 5.4 ABLATIONS

Figure 2a shows localization accuracy for the following ablations: backbone not pretrained (initialized with random weights; black), no sparsity induced during pretraining by a gating mechanism

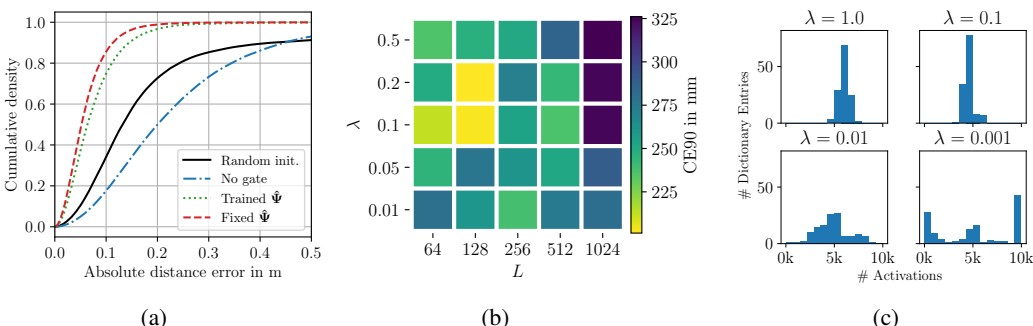

(a)    (b)    (c)

Figure 2: (a) shows the cumulative density of the wireless localization error under ablations, (b) shows the localization accuracy in dependency of the dictionary size $L$ and the sparsity coefficient $\lambda$ and (c) shows the distribution of number of activations on 10000 datapoints.

(blue), and two regular pretraining variants using a trained dictionary (green) and a fixed dictionary corresponding to the theoretical channel model (red). Both regular pretraining cases achieve a very high accuracy of CE90 $\geq 0.145$. The learned dictionary incurs a minor CE90 degradation of $0.003\,\mathrm{m}$ versus the fixed dictionary, so it remains competitive and is suitable when system configurations are unknown. SpaRTran reduces CE90 by $66.5\,\%$ relative to a randomly initialized backbone, demonstrating its effectiveness. Removing the sparsity-inducing gating worsens performance relative to the random initialization, underscoring the critical role of the sparsity assumption.

Figure 2b shows the CE90 localization accuracy w.r.t. dictionary size $L$ and sparsity coefficient $\lambda$. Here the best accuracy is achieved with a dictionary size of $L = 128$ and high levels of sparsity $0.1 \leq \lambda \leq 0.2$. This backs up the claim that forcing the model to express the signal with as little atomic components as possible results in better representations. However, the accuracy reduces with the highest tested $\lambda$. This is caused by a systematic underestimation of the nonzero magnitudes caused by the strong sparsity penalty, an effect known as shrinkage (Wright & Sharkey, 2024). While our gated SAE design mitigates shrinking (Rajamanoharan et al., 2024), it still appears at high $\lambda$. Thus, $\lambda$ balances a tradeoff between sparsity and expressiveness of the representations. It is crucial to select an large enough dictionary size $L$, i.e. dimensionality of the latent, in order to obtain a high reconstruction fidelity. However, in Figure 2b it is noticeable that the accuracy drops for large $L$. This is expected, as a larger dictionary leads to more coherent atoms making it difficult to distinguish which one contributes to the signal (Donoho & Huo, 2001).

Figure 2c shows the distribution of number of activations per atom of a learned dictionary with $L = 128$ dependent on $\lambda$. In general, higher values of $\lambda$ (strong sparsity penalty) lead to a less spread out histogram, i.e. the atoms are activated with equal frequency, indicating a effective diversity of the learned dictionary. It is noticeable that when $\lambda$ is chosen very small ($\lambda = 0.001$), some atoms dominate the representation, being activated constantly while others are stalled, an effect akin to mode collapse.

## 6    CONCLUSION

We presented SpaRTran, an novel unsupervised method for learning task-agnostic radio channel representations based on a gated SAE that integrates a channel model inspired by compressed sensing. This design reflects the inherent sparsity of physical radio channels, resulting in more meaningful and efficient representations. Unlike existing methods, SpaRTran operates on individual radio links rather than full CSI matrices, significantly reducing data acquisition effort and decoupling the model from specific system configurations, making it well-suited for training large, generic foundation models. Finetuned on downstream tasks, SpaRTran outperforms state-of-the-art methods, achieving up to $28\,\%$ reduction in positioning error and a 26%pts increases in the top-1 accuracy on codebook selection for beamforming.

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

## A PROOFS

*Proof of Theorem 1.* Let
$$u := O[f] \in \mathcal{H}$$
then, by definition of $\| \cdot \|_{A_1(\mathcal{H})}$ we have for all $\varepsilon > 0$ a finite set of coefficients $c_i$ such that
$$u = \sum_{i \in I} c_i \varphi, \quad \sum_{i \in I} |c_i| \leq \|u\|_{A_1(\mathcal{H})} + \varepsilon = \|f\|_{A_1^O} + \varepsilon$$
Applying ERC to $u$ we get
$$\sigma_n(u) = \min_{J \subset [N], |J| \leq n} \left\| u - \sum_{j \in J} a_j \varphi_j \right\| \leq C \frac{\|u\|_{A_1(\mathcal{H})}}{\sqrt{n}} \leq C \frac{\|f\|_{A_1^O} + \varepsilon}{\sqrt{n}}$$
by definition, we have a
$$g_n = \sum_{j \in J_n} a_j \varphi_j, \quad |J_n| = n$$
such that
$$\|u - g_n\|_{\mathcal{H}} \leq C \frac{\|f\|_{A_1^O} + \varepsilon}{\sqrt{n}}$$
Finally, set $O^{-1}[g_n] = \tilde{f}_n$ to obtain
$$\|f - \tilde{f}_n\|_{\mathcal{H}} = \|O^{-1}[u - g_n]\|_{\mathcal{H}} \leq \|O^{-1}\| \|[u - g_n]\|_{\mathcal{H}} \leq \|O^{-1}\| C \frac{\|f\|_{A_1^O} + \varepsilon}{\sqrt{n}}$$
Now let $\varepsilon \to 0$ and we obtain our bound
$$\|f - \tilde{f}_n\|_{\mathcal{H}} \leq \|O^{-1}\| C \frac{\|f\|_{A_1^O}}{\sqrt{n}}$$
$\square$

*Proof of Theorem 2.* Let $O$ be a diagonal operator with weights $w_i$ such that $O[f] = \sum w_i c_i \varphi_i$. Let $w_i$ be defined as
$$w_i = \begin{cases} c & j \in S \\ 1 & j \notin S \end{cases}$$
with $c > 1$. Operators with this weighting have inverse norm $\|O^{-1}\| = \max(1/c, 1) = 1$ and
$$\max_{0 \leq i \leq K} \|f_i\|_{A_1^O} = \max_{0 \leq i \leq K} \sum_{j \in S} |a_{i,j}| + \frac{1}{c} \sum_{j \in S} |a_{i,j}| = B + \frac{1}{c} \max_{0 \leq i \leq K} \sum_{j \in S} |a_{i,j}|$$
Since, by assumption $B < R$ we may chose any $c$ large enough such that the residual
$$\frac{R_S}{c} < R - B$$
Specifically, choose
$$c > \frac{R_S}{R - B}$$
then
$$\max_{0 \leq i \leq K} \|f_i\|_{A_1^O} = B + \frac{R_S}{c} < B + (R - B) = R$$
In short, the resulting rate is (nonasymptotically) better than the original one. $\square$

*Proof of Corollary 1.* Let $M_{\text{diag}}$ be the advantage gained by a diagonal scaler. First apply an orthonormal transform $V$ projecting onto $s$ axes (via SVD or polar decomposition), then apply theorem 2. This only adds the burden of back-projecting from $V$ space to the original space, i.e.
$$\max_i \|f_i - f_{i,n}\| \leq \|V^{-1}\| C M_{\text{diag}} / \sqrt{n}$$
which due to orthonormality yields
$$\max_i \|f_i - f_{i,n}\| \leq C M_{\text{diag}} / \sqrt{n}$$
$\square$

## B    EXAMPLE PRECONDITIONING OPERATOR

Assume one wants to solve a Poisson differential equation using the Spectral method on the basis of Legendre Polynomials. Let's define the Poisson equation $-\Delta u(x) = f(x)$ for ground truth $f(x) = \frac{1}{\sqrt{1-x^2}}$ over $x \in (-1, 1)$ with $u(-1) = u(1) = 0$. Approximating $f$ with Legendre polynomials has slow decay for spectral methods due to the endpoint singularities. Preconditioning with Green's operator

$$O := \left(-\frac{d^2}{dx^2}\right)^{-1}$$

Then

$$O[f] = u$$

is smooth over $(-1, 1)$ and in fact has an analytic interior. Specifically legendre coefficients in $f$ decay on the order of $n^{-1}$ due to the singularity, while coefficients in $u$ decay exponentially. For us this would mean

$$\|f\|_{A_1} := \inf \left\{ \sum_i |c_i| \mid f = \sum_i c_i \varphi_i \right\}$$

while

$$\|f\|_{A_1^O} = \|O[f]\|_{A_1} = \|u\|_{A_1}$$

by the convergence rate we get

$$\|f\|_{A_1^O} << \|f\|_{A_1}$$

The norm

$$\|O^{-1}\| = \| - \frac{d^2}{dx^2}\|$$

over $H^2 \cap H_0^1$ - The second order sobolev space cut with the first-order subspace which vanishes on the boundary - is a fixed constant.

## C    EXPERIMENTAL SETUP: BASELINES & DATASETS

### C.1    BASELINES

Table 4: Comparison of transformer hyperparameter.

| Method | $N_{latent}$ | $N_{hidden}$ | $N_{heads}$ | $N_{blocks}$ | #param |
|---|---|---|---|---|---|
| Masking | 512 | 1024 | 8 | 3 | 8.7 M |
| SwiT | 384 | 384 | 1 | 1 | 4.0 M |
| LWM | 64 | 256 | 1 | 12 | 1.3 M |
| SpaRTran (ours) | 512 | 1024 | 8 | 1 | 2.6 M |

**SWiT**: Salihu et al. (2024) propose a joint embedding-based approach Grill et al. (2020) called self-supervised wireless transformer (SWiT), that predicts the output of a momentum encoder, given different augmented views of the same input signal. The aim is to learn representations that are invariant to six randomly selected augmentations, diversifying the views.

**WiT**: Salihu et al. (2022) employs a compact TF model consisting of a single encoder block with single-head attention that is trained end-to-end in a supervised manner.

**Masking**: Ott et al. (2024) introduce a predictive objective for learning FP representations, in which masked portions of the input signal are reconstructed. During training, up to 50% of the input fingerprint is removed, forcing the model to learn spatiotemporal correlations between the multipath components (MPCs).

**LWM**: Alikhani et al. (2024) also use a masking strategy to pre-train the TF model. Here multiple patches of 15 time steps, of the input signal are gathered to be represented to the network as a token. They randomly select 15% of patches and, for those, replace 80% with a uniform MASK vector, 10% with random noise, and leave 10% unchanged.

Table 5: Comparison of dataset configurations.

| Dataset | $f_c$ [GHz] | $W$ [MHz] | $N_r$ | Area |
|---------|-------------|-----------|-------|------|
| KUL | 2.61 | 20 | 64 | $3\,\mathrm{m} \times 3\,\mathrm{m}$ |
| FH-IIS | 3.7 | 100 | 6 | $40\,\mathrm{m} \times 30\,\mathrm{m}$ |
| DeepMIMO | 3.5 | 20 | 32 | $>1\,\mathrm{km}^2$ |

## C.2 DATASETS

**KUL Dataset** Bast et al. (2020): The dataset comprises four antenna configurations: distributed antennas (DIS-LoS), a uniform linear array (ULA-LoS), and a uniform rectangular array under both LoS (URA-LoS) and nLoS conditions (URA-nLoS). Each configuration contains 252,004 CSI samples with recording positions arranged in a grid-pattern with 5 mm distance. The channels are measured at 20 MHz bandwidth. We split the dataset randomly into 70 % for training, 10 % for validation, and 20 % for testing.

**FH-IIS Dataset** Stahlke et al. (2024): This dataset contains CIR fingerprints collected using a 5G-FR1-compatible software-defined radio system (DL-PRS reference signal) with a bandwidth of 100 MHz. The recorded environments resembles a industrial hall featuring tall metal shelves, and a narrow corridor with large walls that introduce signal blockages and complex multipath propagation. The CSI is captured along a random walking trajectory of a person at a sampling rate of 6.6 Hz. Six base-stations are distributed along the perimeter of the localization area. The split sizes are as follows: 566,589 training, 141,639 validation and 593,022 test samples.

**DeepMIMO Dataset** Alkhateeb (2019): This dataset is a synthetic dataset generated by a ray-tracing engine. We configure it to simulate a uniform linear array with 32 antennas at 20 MHz bandwidth, and 3.5 Ghz carrier frequency. We split the dataset into a pretraining and finetuning subset including 15 scenarios (*O1, Boston5G, ASU Campus, New York, Los Angeles, Chicago, Houston, Phoenix, Philadelphia, Miami, Dallas, San Francisco, Austin, Columbus, Seattle*) and 6 scenarios (*Denver, Fort Worth, Oklahoma, Indianapolis, Santa Clara, San Diego*) respectively. This yields the following split sizes: pretraining — 540,272 training and 135,075 validation samples; finetuning — 10,385 training, 1,481 validation, and 1,481 test samples.

We normalize the received signal strength of the FH-IIS and KUL datasets using a global normalization factor. This preserves the relative signal strength differences within the channel measurements, a crucial property for the downstream task of wireless localization. In contrast, DeepMIMO is normalized per CSI matrix to reduce sensitivity to outliers from large peaks that occur at short transmitter–receiver distances (signal strength scales roughly with $1/d^2$).

## D LLM DISCLOSURE

We used large language models (LLMs) to assist with drafting and polishing the manuscript and for literature retrieval/discovery (e.g., to identify related work). Specifically OpenAI GPT-4o and OpenAI GPT-5 were used to rephrase sentences for clarity and conciseness as well as search and summarize candidate related works. All LLM-generated text and suggested citations were reviewed, edited, and validated by the authors; final responsibility for the content, interpretations, and citation accuracy rests with the authors.

