# OpenReview forum: "Simplicity is Key: An Unsupervised Pretraining Approach for Sparse Radio Channels"
_ICLR.cc/2026/Conference — ICLR 2026 Conference Withdrawn Submission_

### Official Review · Reviewer_8QV3 · 2025-10-24

**Soundness:** 2
**Presentation:** 1
**Contribution:** 2
**Rating:** 4
**Confidence:** 2

**Summary:**

The manuscript proposes a transformer based approach for sparse presentation of radio channels.

**Strengths:**

Results looks good.

**Weaknesses:**

Even I have some background in physical layer wireless processing and machine learning, I have some hard time to understand that what is happening the paper. I can understand that the motivation is to find a sparse representations for the channel in terms of a directionally but, two major questions which remains after the manuscript are why I would do that and how I would apply the approach in practice? The paper seems to be a mix of results from previous researches with some specific terminology from research areas which makes it difficult to understand their study. Some concrete examples are in Questions.

**Questions:**

- Section 3.1 is quite theoretical and therefore maybe not easiest to follow. E.g. they say that “One can
sacrifice a more irrelevant subset $\bar S$ in favor of modeling a relevant subset S with higher weight.", I didn't really get the point of that? Also "dense rotation and diagonal scaling" in the latest paragraph seems bit out of context and what is the basis set which is generally infinite? Also there are quite many notations such as $a_{i,j}$ and $f_{i,j}$.  Also, they start with "We treat the incoming signal as a time-dependent function f(t)", but what is this incoming signal actually is? In radio concept, the incoming signal would perhaps be the received signal $y(t)$, but I would assume that they are not aiming to have a sparse presentations for that. Probable it is the channel, but "incoming signal" as quite misleading as channel is needs to be measured indirectly in practice.  (you could just say channel of use notation $h$ instead of $f$).
- Section 4: "Our approach employs an encoder that generates a latent representation $z$ and a decoder that reconstructs the input signal  on z". As the background of the method in this point is still quite fuzzy, I am wondering that how this encoder is meant to be used eventually?
- Section 4: "three time steps of the CIR to an input token $e_m$ ...", does this implicitly assume that the input is always at least three consecutive time steps of CIR? (As intended use of the approach is this point is still quite fuzzy to me, I am wondering how viable this setup this in practice, as often there are no the consecutive time steps of channel measurements especially if this is done using pilots)
- Section 4.1: As $\hat x$ is the encoder output, shouldn't it be e.g. $\hat z$?
- Section 4.1: I am not familiar "active atoms" terminology, but this refers to is nonzero in the sparse representations?

---

### Official Review · Reviewer_fSpF · 2025-11-04

**Soundness:** 3
**Presentation:** 3
**Contribution:** 3
**Rating:** 6
**Confidence:** 3

**Summary:**

The paper proposes SpaRTran, an unsupervised pretraining method for wireless channels that builds a physics-aligned, sparse representation of radio signals. It uses a gated sparse autoencoder plus a learned dictionary of “atoms” to reconstruct signals, encouraging simple codes that reflect multipath sparsity; a phase head models complex values.

SpaRTran is pretrained only on single-link measurements (system-agnostic), then fine-tuned for downstream tasks like indoor localization (fingerprinting) and beamforming. Across several datasets, the approach yields large gains—reporting up to 85% error reduction on fingerprinting—while requiring less pretraining effort than prior SSL baselines

**Strengths:**

- Modeling channels as sparse sums of path atoms, then learning a dictionary and complex phases, is well-matched to multipath propagation and potentially avoids mismatches seen in generic SSL.

- Clear architectural choices. The gate–magnitude decoupling (plus auxiliary reconstruction through the gate) is thoughtful.

- System-agnostic pretraining. Training on single-channel measurements (not full CSI) makes the foundation model less tied to a specific antenna configuration and reduces labeling/collection burden; later, CSI-level heads are added.

- Empirical gains for fingerprinting and beamforming.

- Useful ablations.

**Weaknesses:**

- I don't fully understand the motivation for using a sparse representation. If you can learn an embedding and a dictionary, then can't you get small embeddings (as opposed to sparse but large embeddings) which can make fine-tuning for beamforming more practical?

- I don't understand the purpose of Theorem 1 & 2. Specifically, if your main contributions are the gated sparse-autoencoder, then the operators constructed by hand in the Theorems have no connection to your actual algorithm.

- The algorithm seems to struggle when the number of labelled samples is small. Do self-supervised techniques have the same problem? What's a realistic setting for the percentage of labelled samples?

- The “fixed vs learned dictionary” finding is promising, but practical guidance on dictionary size vs sampling bandwidth or phase noise is limited; no sensitivity study to SNR or carrier frequency offsets.

**Questions:**

Mainly asked them in the weaknesses section

---

### Official Review · Reviewer_DDDr · 2025-11-07

**Soundness:** 2
**Presentation:** 1
**Contribution:** 1
**Rating:** 2
**Confidence:** 4

**Summary:**

This paper applies the gated SAE model, with modifications to handle complex-valued input/output, to discrete time domain channel impulse responses. The problem addressed in this paper is to obtain task-agnostic channel representations useful for downstream tasks by learning to reconstruct sparse representations of the channel vector in the form of a combination of a subset of overdetermined dictionary elements. The authors utilize the fact that a generic multipath channel, modeled with each path's associated complex gain, delay, and phase, can be expressed using only a small subset of an overdetermined set (a dictionary of vectors). After the pretraining with the sparse reconstruction objective is completed, a small ResNet model with 1D convolutions attached to the pretrained encoder is fine-tuned on wireless localization and beamforming tasks with comparisons against baselines.

**Strengths:**

One strength of the solution is that it is based on a system-agnostic channel assumption that enables its use across different communication systems. The theoretical analysis in the paper is strong and motivates the sparsity assumption of time domain wireless channels. Compared to baselines, SpaRTran achieves superior performance in both downstream tasks across different datasets.

**Weaknesses:**

- This work demonstrates limited novelty, as the authors simply extend the gated SAE to handle complex-valued inputs and replace the ReLU activation with LeakyReLU. A major limitation of the work is the encoder's awareness of spatial characteristics. The authors try to circumvent that problem by learning those spatial characteristics separately for each task from scratch by using a ResNet with 1D convolutions added on top of the pretrained encoder. However, this requires processing of every pair of tx-rx elements sequentially (i.e., hundreds of forward passes are needed for very large antenna arrays), which renders the solution infeasible for real-time operability, which is one of the most important requirements in the wireless domain.

- Another major limitation of the paper is the weakness in its presentation, which makes the paper difficult to understand and follow for the readers. A small subset of my observations is detailed below as well as my other comments.

- There are some typos in the introduction (and in other sections). In some places, the language is not clear and is difficult to follow. Some of the sentences have vague meanings. There are also frequent punctuation errors. It is perfectly fine to make typos here and there, but this paper contains way too many of them, making the manuscript hard to read and understand. Therefore, the overall presentation is weak. Some examples (numbered from 1 to 12) from the abstract and the first paragraph only:

1) 28 % should be 28%.

2) 26%pts should be 26%. What is pts? Is it short for percentage points?

3) The claim that “pretraining models solely on individual channel measurements makes it system-agnostic and more versatile” is vague. What is an individual channel measurement?

4) “phase-rotates” and “Doppler-shifts” are used as verbs (line 30), which sounds informal.

5) Missing comma at the end of line 30 (between the words influence and minimizing).

6) Missing comma between the words impairment and leading (line 33).

7) “Pattern” should be plural (line 35).

8) “An” should be “a” (line 37).

9) The authors should use parentheses for the reference in line 38.

10) There are some claims in the introduction that need to be corrected or supported in a stronger way.

11) The claim that “in the past CSI was used to equalize channels and that now CSI is being reinterpreted as a spatiotemporal information source” is not very accurate. CSI is still being used for channel equalization in various systems and is still very relevant in that regard. The authors should back up such an atypical claim with references. Also, the authors attribute the reinterpretation of CSI to the rise of deep learning. Yet, CSI’s use for beamforming predates deep learning’s adoption in wireless domain. Even most current deployments rely on classical signal processing, not deep learning.

12) The claim that “in vision, separating class representations is sensible, but CSI measurements vary smoothly across space and thus exhibit subtler relationships” is interesting. Firstly, natural images also exhibit smoothness and structure (i.e., they lie on low-dimensional manifolds). Secondly, what is the relationship between the separability of classes and smoothness? One can separate CSI samples based on class if the samples are given meaningful labels such as the best beam index from a predefined codebook.

- The related work section presents as a limitation the fact that most papers on SSL and unsupervised learning for wireless borrow methods and techniques from language and image domains. Yet, the proposed solution is based on the Gated SAE framework, developed for interpretability of LLMs. It is not clarified how this work differentiates from the rest in terms of developing a novel method tailored to wireless.

- The related work section does not clearly motivate why this solution is important and how it addresses the limitations of the earlier work developed for the same problem. A more detailed review of the self-supervised learning for wireless domain would be beneficial to show if/how earlier works do not account for the domain-specific characteristics of wireless but simply adopt methods developed purely for image and language. Also, this method uses time domain CIR as the input while most other cited works and baselines use frequency domain channel response. What are the advantages and disadvantages of the authors’ choice of using time domain CIR? Those details and comparisons are missing, leaving the motivation for the design choices and the need for this solution blank.

-  A more detailed analysis of the model behavior (i.e., effect of the complexity of the downstream model, effect of model dimension, number of attention heads, and transformer depth, etc.) and interpretability of the learned features would strengthen this paper. The overall picture here looks like the authors have applied yet another method developed for another domain to wireless, which in the first place is what is criticized by the authors.

-  The lack of spatial understanding introduces a dependency on the expressivity of the downstream model, which is a small ResNet with 1D convolutions here. The authors should analyze the performance under different downstream models as it is part of the design process unlike other baselines where simpler downstream models may be sufficient due to the spatial understanding already embedded in the encoder representations.

- The datasets in this work use a carrier frequency around 3 GHz. However, the characteristics of the channel vary greatly with the carrier frequency. An evaluation on higher frequencies (mmWave frequencies) would strengthen this work.

**Questions:**

- The pretraining and finetuning steps should be included in detail for SpaRTran and baselines. For example, is the same ResNet attached to all the baseline encoders, or you have simply used linear predictors for the other baselines, .... Under which settings have those baselines been implemented, trained, and finetuned?

- SpaRTran encodes the CIR between a pair of transmitting and a receiver element. For a MIMO or a MISO system, this means that we must encode the channel for every pair of elements and then use another network (the authors used a small ResNet with 1D convolutions) to learn the spatial characteristics and dependencies, which are fundamentally important for both of the downstream tasks considered in this work. Other baselines already consider the spatial correlations with a single forward pass, which makes them orders of magnitude more efficient for systems with large antenna arrays. When the antenna arrays are very large, which is the case for modern communications, the proposed method has an important scalability problem. How can this be addressed?

- Why is the work in https://arxiv.org/abs/2505.09160 cited as a previous work but not considered as a baseline? It seems to study a similar problem and even has evaluations on the same downstream task on the same dataset.

- How is compressed sensing utilized in this solution? The gated SAE framework, which the authors adopted here, already enforces sparsity, and the authors do not propose a fundamental modification to Gated SAEs. In the original gated SAE paper, the sparsity is not presented as an application of compressed sensing. This paper, on the other hand, uses the same gated SAE method yet frames it as a novel application of CS, being the first CS application to design of unsupervised learning, which may be an overstatement.

- Please also see some of the questions in the Weaknesses section.

---

### Official Review · Reviewer_Dhgi · 2025-11-11

**Soundness:** 2
**Presentation:** 2
**Contribution:** 2
**Rating:** 2
**Confidence:** 4

**Summary:**

This paper proposes an unsupervised pretraining method for wireless channels named SpaRTran. The technique uses concepts from compressed sensing, gated autoencoders, and dictionary learning to learn representations. The encoder uses a transformer, a gated sparse autoencoder, and a learned dictionary. The main idea is to use a sparse representation of the wireless multipath channels to learn representations between point-to-point wireless links. The pretrained model is then fine-tuned for downstream tasks, including fingerprinting-based localization and beamforming codebook selection. Experiments show improvements over the existing baselines.

**Strengths:**

1. The proposed method builds on a well-established sparse multipath channel model and applies a compressed sensing-based method. The formulation is well motivated and captures the nature of wireless channels. This makes the architecture design of the learning model more interpretable.

2. The gain over the existing methods is illustrated with empirical experiments over two different tasks and several datasets.

**Weaknesses:**

1. The novelty is obviously overclaimed; applying compressed sensing to the design of unsupervised pretraining is not new.
A brief search over the existing literature I found several works that have explored this idea.

e.g. Uncertainty Autoencoders: Learning Compressed Representations via Variational Information Maximization, AISTATS 2019.
What is novel here is the compressed sensing-inspired unsupervised pretraining method for CSI representation in wireless channels.

2. The theory section (3.1) is unclear and poorly integrated. This part introduces an abstract operator approximation framework with some fairly strong assumptions, but the theorem's symbols are not mapped to the original model, and the connection to the previous sections is not well explained. Moreover, there is no empirical verification of the theoretical results in the learned dictionary. These together make the whole part seem irrelevant to the main paper.

3. While the main paper is talking about a learnable dictionary, the learned one did not outperform the fixed one. The construction of the fixed dictionary is not explicitly described, but it just appears in the experiment part without explanations.

4. The method itself is quite specific for the wireless domain. The methods are not new and do not provide generalizable insights to other areas of machine learning. The paper reads as a well-engineered domain-specific method, but the method is not tested in other domains that also use wave representations. This makes the paper a questionable contribution to the ICLR conference.

5. The experiments are limited to synthetic ray-tracing generated datasets in the beamforming codebook selection task.

6. Details about the experiments are missing, including the model size, data quantity, pretraining time, etc. This makes one to question the fairness of the experiments.

**Questions:**

1. The paper primarily pertains to point-to-point wireless links.  However, in the real world, most communication systems are equipped with multiple antennas. Ignoring the correlation between different antennas would probably result in a loss of performance. Why not start with a more general model that pretrains on MIMO channels?

2. The ablation study on L and $\lambda$ can probably be enriched by showing the trend of changes in the performance of the downstream tasks. For different tasks, is there an optimal $\lambda$, and what will be the corresponding sparsity?

3. There are some cases where SWiT outperforms the proposed method. Is there a better explanation for that?

---

### Note · Authors · 2025-11-13

**Comment:**

Thank you very much for the valuable feedback provided by the reviewers. We greatly appreciate the time and effort invested in evaluating our submission.

After careful consideration, we have decided to withdraw our paper. Given the scope and depth of the feedback, we do not believe that we will be able to adequately address all the comments before the deadline. We hope to revise the manuscript thoroughly and submit it to another conference in the future.

Thank you again for your consideration and support.

**Withdrawal Confirmation:**

I have read and agree with the venue's withdrawal policy on behalf of myself and my co-authors.